# Temporal distribution shifts of Chum salmon (*Oncorhynchus keta*) with sea surface temperature changes at their southern limit in the North Pacific

**Beom-Sik Kim**[1], **Hae Kun Jung**[2], **Jong Won Park**[1], **Ju Kyoung Kim**[3], **Chung Il Lee**[1]*

**1** Department of Marine Ecology and Environment, Gangneung-Wonju National University, Gangneung, Gangwon-do, Republic of Korea, **2** Fisheries Resources and Environment Research Division, East Sea Fisheries Research Institute, National Institute of Fisheries Science, Gangneung, Gangwon-do, Republic of Korea, **3** Aquatic Living Resources Center of South Sea, Korea Fisheries Resources Agency, Wando, Jeollanam-do, Republic of Korea

* leeci@gwnu.ac.kr

## Abstract

Understanding the responses of marine organisms to environmental changes at their distribution limits is crucial for predicting climate-change associated habitat changes. This study analyzed the effect of sea surface temperature (SST) on the temporal distribution of Chum salmon (*Oncorhynchus keta*) in the eastern and southern coastal waters of Korea (ESCK) and on the southern limit of their distribution in the North Pacific. The temporal distribution of Chum in the ESCK and adjacent rivers was statistically compared based on three SST types (T1–T3). Chum were first caught in the northern and then in the southern area, with riverine migration occurring faster in the south than in the north. These migration patterns did not change with SST type. There was no significant difference in the coastal arrival timing of the Chum between T1 and T3, which respectively represented the entire region cooling either rapidly or slowly compared to an average year. In T2, in which the north cooled rapidly and the south cooled slowly, the coastal arrival timing was approximately 4 days earlier compared to T1 and T3. Moreover, as the SST type shifted from T1 to T3, the coastal residence time in the north became shorter, while in the south became longer. These findings help us to understand the adaptation strategies of Chum, and to predict changes in their distribution and resources in the North Pacific under climate change.

## Introduction

Climate change affects marine ecosystems similar to agricultural, terrain, freshwater ecosystems [1–3]. Recently, marine ecosystems have been affected by environmental changes due to global warming, such as the poleward shift of subtropical waters and strengthened stratification [4,5]. These changes have caused the suitable temperature habitats of marine organisms to shift poleward [5,6], increased thermal stress [7–9] and enhanced metabolic activity [10–13]. These shifts have led to changes in the ecological characteristics of marine life,

**Data availability statement:** This study includes processed data derived from third-party sources, specifically the "Amount of Catch in the Coastal Water" and "Amount of Catch in the River." These data are not publicly available due to third-party restrictions. Interested researchers may access these data by contacting the respective organizations. The contact information is as follows: Korea Fisheries Resources Agency at logonkjk@fira.or.kr. Appropriate permissions and approval may be required to gain access. All other data generated or analyzed during this study are publicly available in the following GitHub repository: https://github.com/kbs8670/KR_Salmon_Timing.git.

**Funding:** This research was supported by the Basic Science Research Program through the National Research Foundation of Korea (NRF) funded by the Ministry of Education (RS-2024-00412976) awarded to BSK, the National Institute of Fisheries Science, Ministry of Oceans and Fisheries (R2025009) awarded to HKJ, and the Korea Institute of Marine Science & Technology Promotion (KIMST) funded by the Ministry of Oceans and Fisheries (20220558) awarded to CIL. The funders had no role in study design, data collection and analysis, decision to publish, or preparation of the manuscript.

**Competing interests:** The authors have declared that no competing interests exist.

such as their distribution, phenology, movement, and behavioral traits, as they adapt to the changing environmental conditions. Specifically, they are expanding their distribution range poleward by an average of 30.6 km per decade and increasing their average spawning and blooming period by 4.4 days per decade [6,146,14]. Anadromous fishes, such as Salmonidae, must respond and adapt to environmental changes in both freshwater and marine ecosystems throughout their life cycles. During their extensive migration, anadromous fish are frequently exposed to environments that vary in intensity with climate change, such as the Kuroshio and Oyashio currents [15,16]. In the North Pacific, anadromous fishes have reacted sensitively to recent environmental changes, leading to increased abundance of pink salmon in the northern Bering Sea, improved survival rates of Salmonidae in Alaska, decreased survival rates of Pink, Chum, and Sockeye salmon in Washington and British Columbia [17,18], and the early spawning and redistribution of Chinook salmon in the northeastern Pacific [15].

Chum salmon (*Oncorhynchus keta*) is an anadromous fish that plays a critical role in energy transfer between terrestrial, riverine, and marine ecosystems throughout its life. Water temperature is a significant environmental factor that affects the migration timing and distribution of Chum [19–21]. According to the poleward expansion of warm water masses, favorable habitats extend towards the Arctic Ocean [15,22], and warming of these habitats accelerates the growth and maturation of Chum [23,24]. These environmental changes have led to variations in the physiological and ecological characteristics of Chum. For instance, in Alaska, the timing of spawning migrations has advanced by more than two weeks and has become more synchronized across different rivers [22]. There has also been an increase in the number of Chum spawning in the rivers of the Alaskan Arctic [25,26]. Although the responses of Chum to temperature changes have primarily been studied in high-latitude regions, a lack of understanding of these responses along the distributional limits of Chum remains.

Coastal waters are crucial habitats for Chum as they transition between two ecosystems with different environmental characteristics. During spawning migration, when warm water is distributed in the upper coastal water layers, Chum move to deep water layers to control their energy expenditure and maturation rates. They remain in these waters for an extended period and repeatedly move vertically to check the upper water temperature and navigate to their natal river using river signals [27–31]. The amount of time Chum spend in these deeper layers and their vertical movements in coastal waters influence the timing of their riverine migration [28,32–34]. The timing of this migration affects the hatching time and the growth environment of subsequent generations, thereby affecting their survival [29,30,35]. Countries bordering the North Pacific have long recognized the importance of understanding Chum response strategies to coastal environmental changes to predict changes in their distribution, resource abundance, and return patterns. Consequently, extensive research has been conducted on the relationships between coastal environmental changes, such as changes in coastal water temperature and riverine water inflow, and the movement and distribution patterns of Chum [36–40].

The eastern and southern coastal waters of Korea (ESCK) represent the southernmost distribution limits of Chum in the North Pacific. Uniquely in the ESCK, Chum, which prefers cold water and avoids radical environmental changes, migrates to various regions for spawning. These regions include the eastern coastal waters of Korea (ECK), where the boundaries between the warm and cold-water masses are distributed, and the southern coastal waters of Korea (SCK), where warm water masses influence even the deeper layers [41–44]. Because Chum must migrate from the coast to rivers through the upper coastal layers, the continuous return of Chum to the ESCK and its adjacent rivers, where warm water masses are distributed in the upper layers, incites curiosity. The ECK is the region within the ESCK where Chum is most abundantly distributed [30,45]. The ECK is strongly influenced by the Tsushima Warm

Current in the upper layers, has cold water masses persistently distributed in the deeper layers, and features a subpolar front around 37°–38° latitude [46,47]. This creates a diverse set of environmental conditions within a relatively narrow space compared to higher-latitude regions, such as Alaska and the Kamchatka Peninsula. The high concentration of Chum in the ECK is believed to be related to its unique environmental characteristics, specifically the year-round presence of cold-water masses (<15 °C) at depths of 30–50 m (S1 Fig).

The environmental conditions in the ESCK have changed with climate change, including an increase in sea surface temperature (SST) by about 1.5 °C compared to that in the past [48], strengthened stratification [49], and increasing frequency of events such as marine heat waves [50]. Accordingly, these changes have led to an environment that is increasingly unfavorable for spawning-migrating Chum. Despite these changes, the number of returning Chum, which significantly declined in the late 1990s, has steadily increased from the early 2000s to the present [51]; (S2 Fig). This raises various questions regarding the adaptive strategies of Chum to environmental changes in the ESCK and potential future changes in their distribution and resource abundance.

Studies on Chum ecology in the ESCK have mainly focused on the ECK and have aimed to identify the environmental factors influencing the migration and behavioral characteristics of spawning-migrating Chum in each region [52,53]. Studies have found that Chum in the ECK migrate to rivers when the SST decreases to below 20 °C, during high tides when the estuarine space expands, and during spring tides when the mixing between the upper and lower coastal layers is enhanced [44]. These findings suggest that the migration and behavioral patterns of Chum vary according to vertical temperature distribution and tidal changes. However, spawning-migratory Chum must encounter diverse and changing environmental conditions in the ESCK to allow them to navigate to their natal river, which suggests that their responses to environmental changes may vary depending on the specific spawning sites they return to.

In this study, SST was considered the primary environmental factor influencing the migration of spawning-migratory Chum. We hypothesized that SST changes influence not only the timing of coastal arrival but also the duration of delay in coastal waters before river entry, thereby affecting the overall migration timing and patterns in the ESCK and its adjacent rivers. To elucidate this, the distribution of SST in the ESCK was divided into three types, and the timing of spawning-migratory Chum migrating to the ESCK and its adjacent rivers was compared for each environmental type. Studying Chum responses to marine environmental changes in the ESCK, which displays threshold conditions for Chum, is crucial for understanding their responses and adaptation strategies to the distributional shift of warm water masses. Additionally, this study provides valuable data for predicting future changes in the distribution and resource abundance of Chum in the North Pacific region.

## Materials and methods

### Study area

Chum return to the ECK and SCK in the ESCK between October and December (Fig 1). In the lower reaches of seven rivers in the ECK and one river in the SCK, catching and population checks were conducted almost daily using fishing gear installed for Chum return rate analysis and artificial propagation. Additionally, at the ports in each region, the daily number of individuals caught by the set and gill nets in the ESCK was recorded. Coastal fishing activities were also conducted almost daily during the season when Chum return to spawn. It is important to note that fishing activities were not limited to specific locations or vessels; instead, the data were collected across a range of areas within each region, with samples considered to be

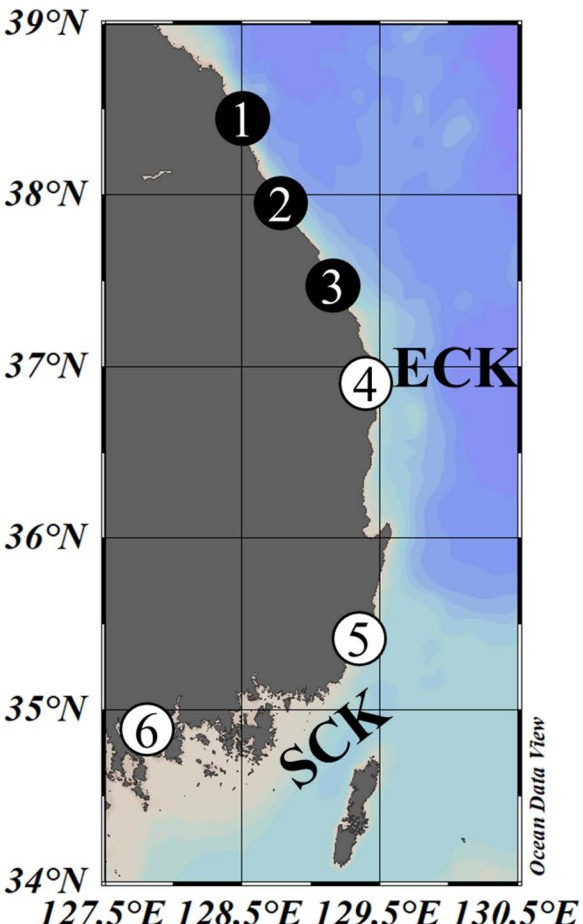

**Fig 1. Map of the study area, showing the eastern coastal waters of Korea (ECK) and southern coastal waters of Korea (SCK), collectively referred to as the eastern and southern coastal waters of Korea (ESCK).** The study area in the ESCK was divided into the northern and southern areas based on latitude 37° N. The black and white circles with numbers indicate the study regions in the northern (CR1–CR3) and southern (CR4–CR6) area, respectively, selected to compare the temporal distribution of Chum. The map was generated using Ocean Data View (ODV) version 5.7.0, available at https://odv.awi.de.

randomly distributed. While fishing efforts might not have occurred consistently at the same locations, this variability was treated as a random error in the dataset.

The study area was divided into the northern and southern area based on latitude 37° N (Fig 1), where the subpolar front is mainly distributed during the spawning-migration season [46,47]. The research regions in each area were selected based on rivers in which Chum-fishing gear had been consistently installed for the longest period and those with the highest return rates (Fig 1). Three regions were chosen for each area: CR1–CR3 for the northern area, and CR4–CR6 for the southern area (Fig 1). The geographic coordinates of each site are as follows: CR1 (38.55°N, 128.41°E), CR2 (37.86°N, 128.85°E), CR3 (37.44°N, 129.15°E), CR4 (36.97°N, 129.41°E), CR5 (35.47°N, 129.39°E), and CR6 (34.95°N, 127.77°E).

## Sea surface temperature distribution types in the ESCK

In the ESCK, Chum migrate from coastal waters to rivers when the SST decreases below 20 °C [44]. The distribution of the SST in the ESCK was categorized into three types based on

the date on which the SST first decreased below 20 °C in each region (S3 Fig). The SST data were obtained from the OSTIA daily SST reanalysis data, covering the period from October 1 to December 31 each year from 2006 to 2018. The geographic coordinates of each region were as follows: CR1 (38.5–38.7°N, 128.4–128.6°E), CR2 (37.8–38.0°N, 128.9–129.0°E), CR3 (37.4–37.6°N, 129.1–129.3°E), CR4 (37.0–37.2°N, 129.5–129.7°E), CR5 (35.4–35.6°N, 129.4–129.6°E), and CR6 (34.9–35.1°N, 127.8–128.0°E). SST distribution type T1 refers to years when the SST across all regions of the ESCK decreased below 20 °C earlier than average year, with the northern and southern area cooling at similar times. Examples include 2006, 2008, and 2018 (S3 Fig, S1 Table). Type T3 represents years when the SST across all regions of the ESCK decreased below 20 °C later than average year, again with the northern and southern area cooling at similar times. Examples include 2007, 2010, 2012, 2014, and 2017 (S3 Fig, S1 Table). In contrast, type T2 is characterized by a significant temporal difference between the northern area (CR1–CR3) and the southern area (CR4–CR6), where the SST in the northern area decreased below 20 °C earlier than in the southern area, regardless of whether the cooling occurred earlier or later than the average year. Examples include 2009, 2011, 2013, 2015, and 2016 (S3 Fig, S1 Table).

## Number of migrating Chum in the ESCK and adjacent rivers

The temporal distribution of Chum migration in the ESCK and adjacent rivers was determined using daily Chum catch data from each port and river along the ESCK from 2006 to 2018 (S4 Fig). The coastal arrival timing was estimated based on the daily number of Chum caught by set and gill nets at ports within 20 km radii of the mouths of the CR1–CR6 rivers. The coastal residence time, which means the time from the coastal arrival timing to the time of entry into the river, was estimated using daily catch data from fishing gear installed in the lower reaches of rivers CR1–CR6.

## Statistical analyses: Linear mixed-effects model

A linear mixed-effects (LME) model analysis was conducted using MATLAB to analyze how the coastal arrival timing and residence time varied according to the SST distribution types. The Akaike information criterion (AIC) was used to select the optimal equation for analyzing coastal arrival timing and residence time (Table 1). The AIC value indicates the fit and complexity of a model equation, with lower AIC values signifying a better fit and lower complexity. The equation with the lowest AIC value was selected as the most appropriate model [24,54]. In the LME analysis, SST distribution type was used as a fixed effect. The region was either used as an interaction term in the fixed effect, used as a random effect, or not used at all. The dependent variables were the coastal arrival timing and residence time. Coastal arrival timing was defined as the median date of migration (MDMT) at the ESCK, determined as the date when more than 50% of the total number of Chum were caught between October and December each year. The coastal residence time was defined as the relative time difference between the MDMT at the ESCK and the MDMT in adjacent rivers.

**Table 1. Akaike Information Criterion (AIC) results of the linear mixed-effects model (LME) for each equation regarding the median date of migration timing (MDMT) of Chum in the ESCK and adjacent rivers.**

| Equation | AIC (Coastal) | AIC (Riverine) |
|---|---|---|
| (1) $MDMT \sim SST\_Types.$ | 439.87 | 237.17 |
| (2) $MDMT \sim SST\_Types + (1 \mid Regions).$ | 426.98 | 236.39 |
| (3) $MDMT \sim SST\_Types * Regions.$ | 431.68 | 233.75 |

Various equations used were as follows: (1) The MDMTs from CR1 to CR6 are almost same, but MDMT varies according to SST distribution type, (2) MDMTs vary across regions, but the magnitude of variation in MDMT across SST distribution types is consistent across all regions (e.g., MDMT gradually shifts later from CR1 to CR6, but under SST distribution type T1, MDMT is consistently 3 days earlier across all regions compared to other types), and (3) MDMTs vary by region and depend on the SST type, i.e., in some regions, MDMT under SST type T1 occurs 3 days later than under T2 and 4 days earlier than under T3, while in other regions, MDMT under T1 occurs 5 days earlier than under T2 and 4 days earlier than under T3 (Table 1).

## Statistical analyses: non-linear regression

Nonlinear regression analysis (NLA) was conducted on the SST and the coastal and riverine migration volume by date in each region to determine how the preferred environment differs when Chum migrates to the ESCK and adjacent rivers based on the SST distribution type. To calculate the mean migration volume by temperature range for each SST distribution type, the number of Chum caught per date, compiled annually for each region, was segmented and accumulated by temperature range. The NLA modeled the relationship between SST and migration volume, determined the temperature ranges at which the migration volumes increased significantly, and analyzed the characteristics of this relationship. Statistical metrics such as mean squared error (MSE) and AIC were used to compare the suitability of various NLA models (polynomial, exponential, and logistic; S2–S5 Tables) and select the optimal regression model [55–57]. The MSE represents the average of the squared differences between the predicted and observed values, with a lower MSE indicating higher prediction accuracy (S4 and S5 Tables). In the NLA, AIC was used in the same way as in LME analysis, considering both the fit and complexity of the model, with lower AIC values indicating better fit and lower complexity (S2–S5 Tables). The polynomial regression model was identified as the optimal model for the NLA based on a comparison of these metrics (S2–S5 Tables).

The model used in this analysis is represented by the equation:

$$y = b_1 + b_2x + b_3x^2 + \in \qquad (A)$$

where, $y$ represents the volume of migrating Chum accumulated in the temperature range, $x$ represents the SST, and $b_1$, $b_2$, and $b_3$ are the coefficients of the intercept, linear, and quadratic terms, respectively. The term $\in$ accounts for the error in the model. Coefficient $b_1$ represents the estimated migration volumes when the SST is zero, providing a baseline migration volume independent of the SST (S4 and S5 Tables). Coefficient $b_2$ represents the change in migration volume with variations in the SST (S4 and S5 Tables); if $b_2$ is positive, the migration volume increases as the SST increases, and conversely, if $b_2$ is negative, the migration volume decreases as the SST increases. The coefficient $b_3$ represents the quadratic relationship between the SST and migration volume, capturing the curvature in the relationship (S4 and S5 Tables). A positive $b_3$ indicates that the change in migration volume increases more rapidly at a higher SST, whereas a negative $b_3$ indicates that the change in migration volume increases more slowly at a higher SST.

## Ethic approval

This study utilizes previously collected fishery data from various institutions for analysis and does not involve the collection or experimentation on live animals. As a result, no additional ethical approval was required.

## Results

### Changes in the temporal distribution patterns of Chum in the ESCK by SST type

In the LME model, Equation (2), which sets the SST distribution type as a fixed effect and the region as a random effect, had the lowest AIC. Therefore, it was optimal among Equations (1)–(3) (Table 1). In the ESCK, the MDMT was earlier in the northern area (CR1–CR3) than in the southern area (CR4–CR6) across all SST types, with a standard deviation of 2.7 days among the regions (Figs 2 and 3). No significant differences were observed between groups T1 and T3. However, in type T2, the MDMT was significantly earlier by approximately 4 days compared to the T1 and T3 types across all regions of the ESCK (p < 0.05; Fig 3).

The NLA results indicate that the polynomial regression model had the lowest AIC in most cases, except for CR6 under T1 conditions (S2 Table). Additionally, the MSE was below 0.002 in most cases, except for CR5 and CR6 at T1, CR6 at T2, and CR5 at T3, which indicated that the polynomial regression model was optimal for this analysis (S2 Table). The $b_1$, $b_2$, and $b_3$ values of the polynomial regression model were statistically significant in most cases, except for CR6 at T1 (S4 Table). This suggests that, except for CR5 and CR6 under T1 conditions, a curvilinear relationship was observed in most cases, where the coastal arrival migration volume peaked within a specific temperature range.

In type T1, the coastal arrival migration volumes were substantial in the northern area (CR1–CR3) in the 15–20 °C range, peaking at around 18–19 °C. In the southern area (CR4–CR6), the migration volumes were substantial in 19–21 °C, peaking at around 20 °C (Fig 4). In type T2, the coastal arrival migration volumes increased notably from 20 °C to 21 °C in the northern area, peaking at around 18 °C. In the southern area, the migration volumes increased notably from 20 °C to 21 °C, peaking at around 21 °C (Fig 4). In type T3, the migration volumes in the northern area were notably high between 18–20 °C, with the highest volumes at around 18 °C. In the southern area, the migration volumes were notably high in the 20–21 °C range, with the highest volumes at around 21 °C (Fig 4).

### Changes in the temporal distribution patterns of Chum, before migrating to rivers adjacent to the ESCK by SST type

In the LME analysis of coastal residence time by SST distribution type for each region of the ESCK, equation (3), which included SST type and region as fixed effects, had the lowest AIC and was therefore optimal among equations (1)–(3) (Table 1). This suggests that Chum coastal residence time varies by region, and that the scale of this variation in response to SST type differs across regions (Figs 5 and 6).

There were statistically significant differences in CR1 by SST distribution type, with coastal residence time being 6.40 days shorter in T2 and 9.60 days shorter in T3 than it was in T1 (p < 0.1, p < 0.05; Fig 5). For CR2, the coastal residence time tended to become shorter from T1 to T3, although the significance of these differences was low (Fig 5). At CR5, the coastal residence time of Chum was nearly the same under T1 and T3, whereas Chum spent an additional 5.00 days in coastal waters before entering the river under T2 (p < 0.1; Fig 5). At CR6, the coastal residence time increased under both T2 and T3 compared to T1, with a more pronounced delay under T2, as Chum spent more time in coastal waters before entering the river (p < 0.05; Fig 5).

Under T1 conditions, Chum spent 10.33, 17.00, and 18.00 fewer days in coastal waters at CR2, CR5, and CR6, respectively, compared to CR1, with statistically significant differences (p < 0.05, p < 0.001; Fig 6). Under T2 conditions, the coastal residence time of Chum at CR2

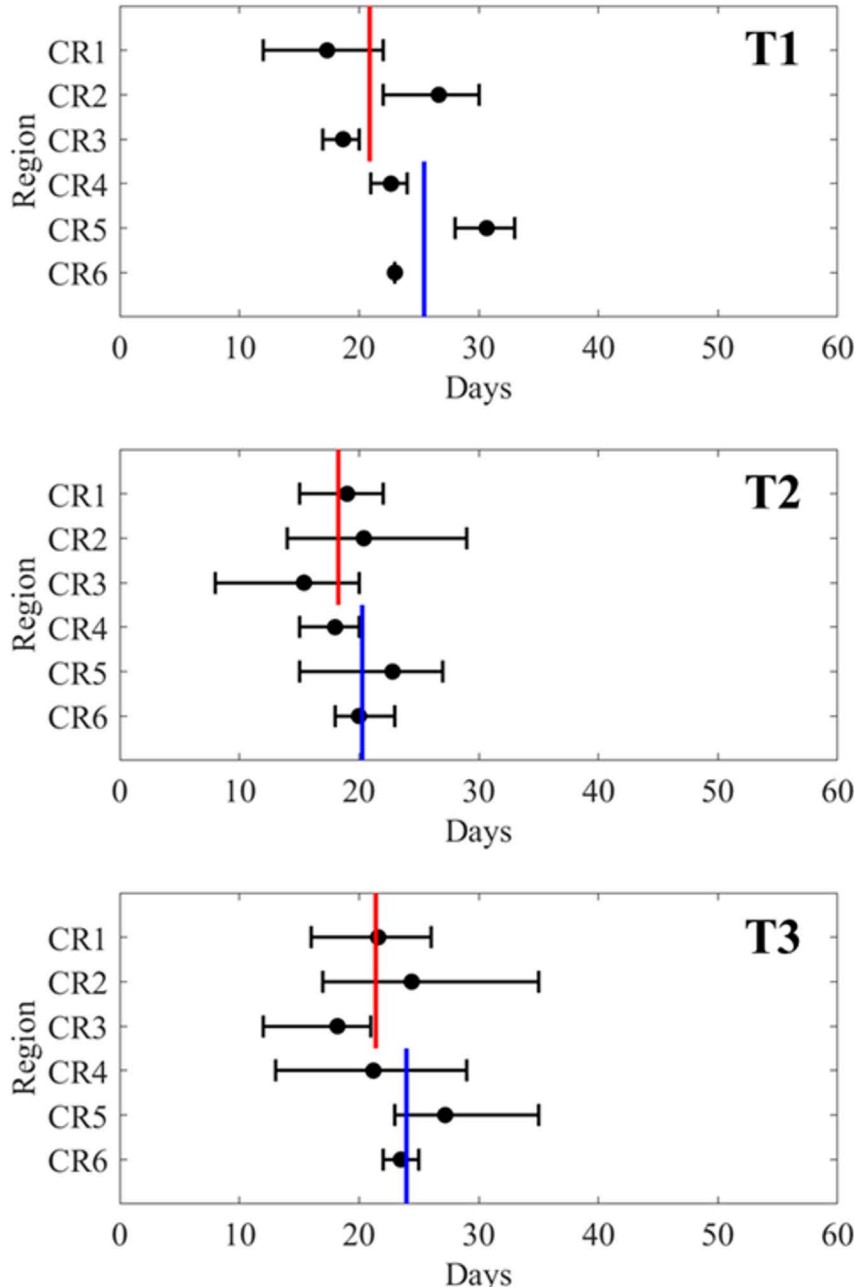

**Fig 2. Average median date of Chum migration timing in each region of the ESCK (CR1–CR6) by SST type (T1–T3).** The black circles with error bars represent the average of the median Chum migration timing for each region by SST type. The red and blue lines indicate the averages of these values for the northern (CR1–CR3) and southern (CR4–CR6) area, respectively.

and CR5 was 5.20 and 5.60 days shorter, respectively, compared to CR1 (p < 0.1), while the coastal residence time at CR6 was almost the same as that at CR1 (p < 0.05; Fig 6). Under T3 conditions, the coastal residence time of Chum at CR2 and CR5 was 3.00 and 7.40 days shorter, respectively, compared to CR1, although the statistical significance was low. The coastal residence time at CR6 was 2.40 days shorter, with statistical significance (p < 0.05; Fig 6).

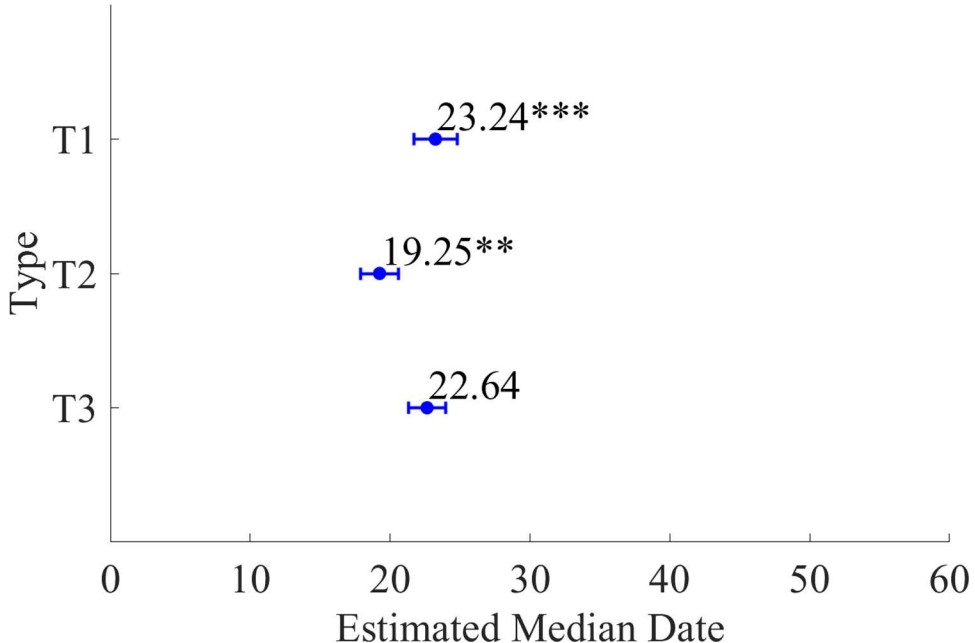

**Fig 3. The estimated median date of migration of the Chum arriving at the ESCK (CR1–CR6) for each SST type (T1–T3), calculated using the linear mixed-effects model.** The blue circles with error bars indicate the estimated median date of Chum migration timing. '**' indicates that p < 0.05.

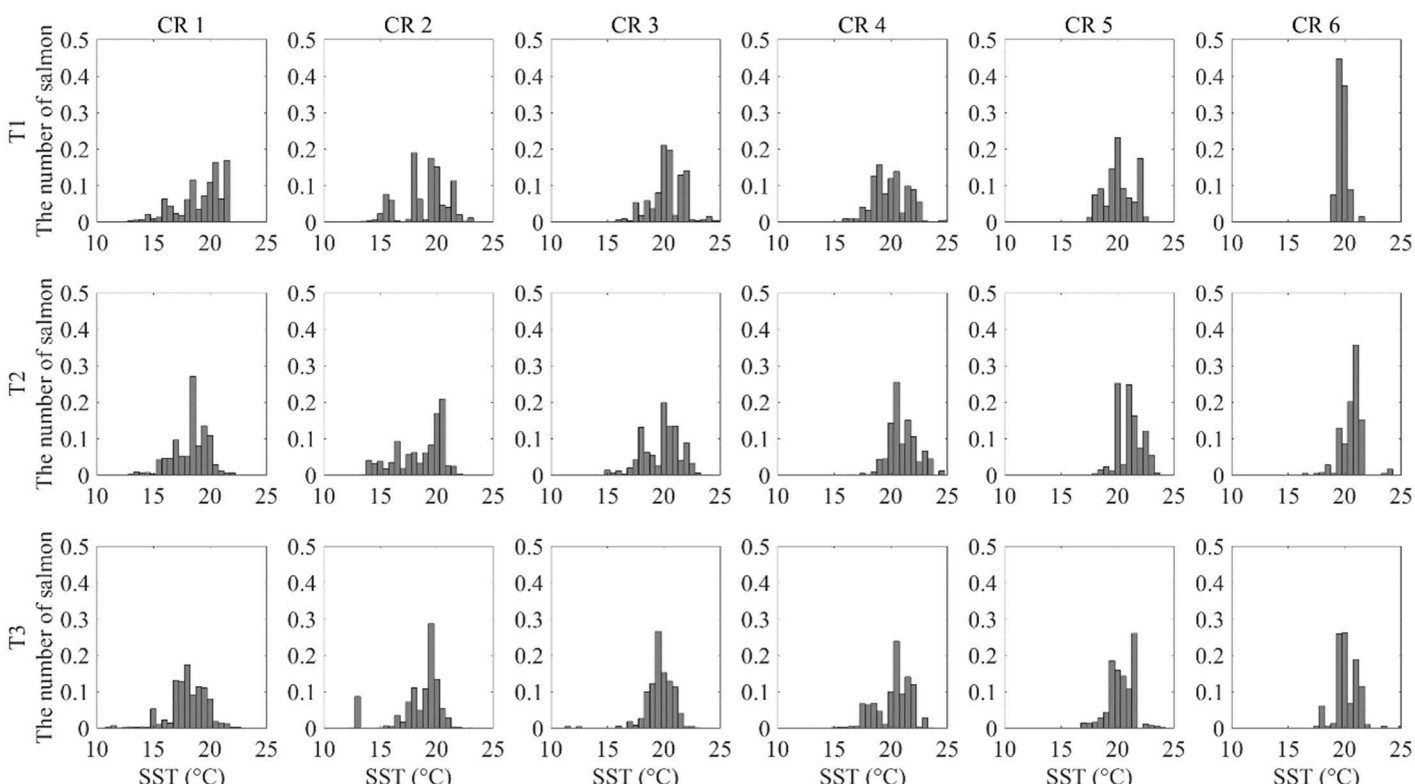

**Fig 4. The migration volume of Chum arriving in the ESCK (CR1–CR6) in each water-temperature range by the type of sea surface temperature distribution (T1–T3).** The x-axis represents the water temperature, while the y-axis represents the migration volume of Chum that arrived at the ESCK within that water-temperature range.

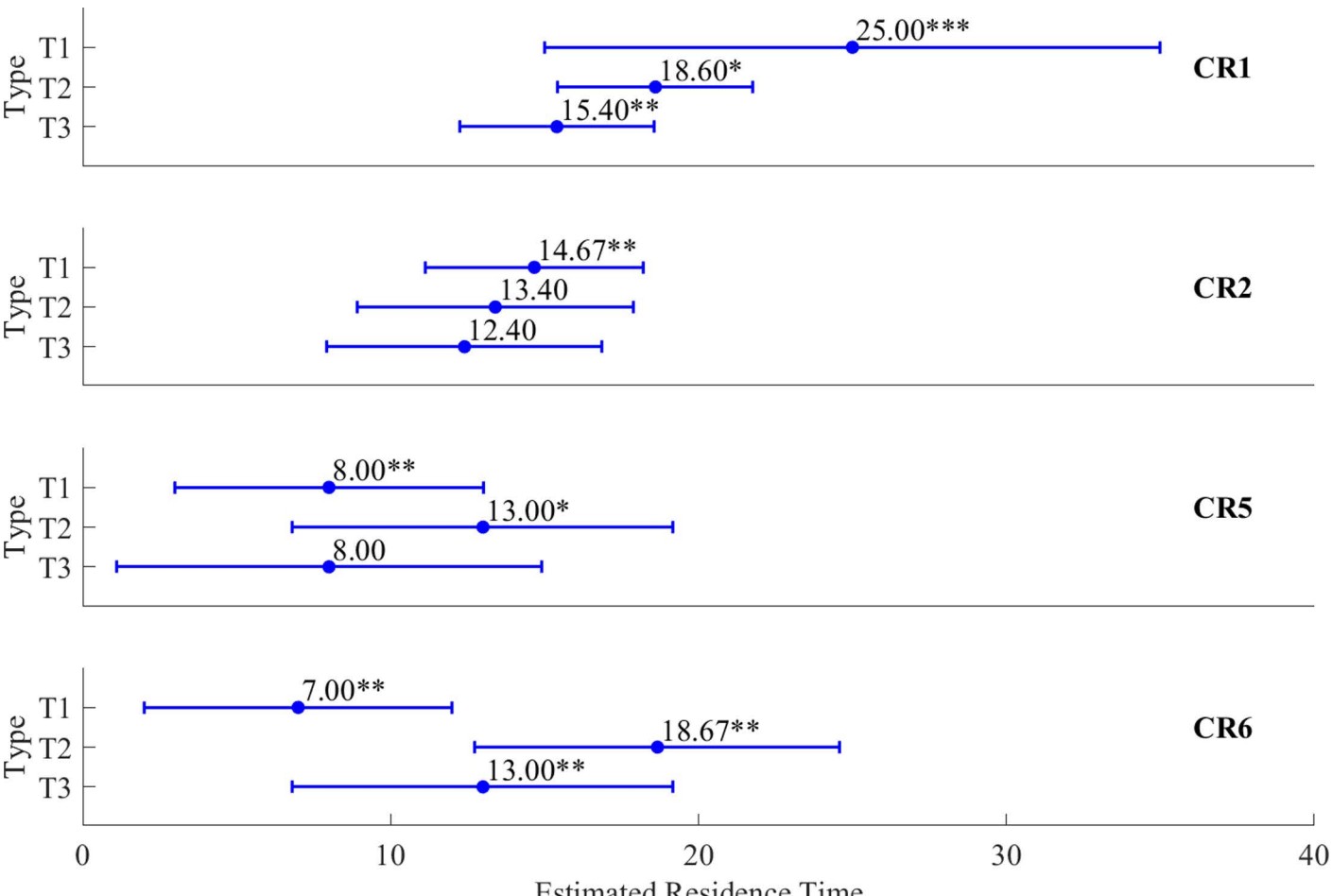

**Fig 5. Comparison of differences in coastal residence time of Chum by SST type (T1–T3) within each region (CR1, CR2, CR5, CR6) based on a linear mixed-effects model.** The blue circles with error bars indicate the estimated residence time of Chum. '*' indicates that $p < 0.01$. '**' indicates that $p < 0.05$. '***' indicates that $p < 0.001$.

Overall, Chum tended to spend less time in coastal waters near the southern rivers (CR5–CR6) than near the northern rivers (CR1–CR2). Although the coastal residence time patterns between the northern and southern rivers remained unchanged, prolonged periods of warm water at the sea surface caused Chum to reduce their residence time near the northern rivers and increase it near the southern rivers, thereby narrowing the residence time differences between the northern and southern rivers.

According to the NLA, the polynomial regression had the lowest AIC in most cases except for the CR2 under T1 conditions and the CR1 under T2 conditions (S3 Table), and the MSE was below 0.002 in most cases except for the CR1 across T1–T3 (S3 Table). In the polynomial regression model, the constants b1, b2, and b3 were statistically significant in most cases, except for the CR1, CR2, and CR5 under T1 conditions, the CR1 under T2 conditions, and the CR5 under T3 conditions (S5 Table). This indicates that the riverine migration volumes could be maximized within a specific temperature range.

In type T1, the Chum riverine migration volume increased in the northern area rivers at 13–14 °C and 17–18 °C, while in the southern area rivers, volumes peaked at around 19–20 °C (Fig 7). In type T2, the riverine migration volume increased at 17–18 °C in the northern area

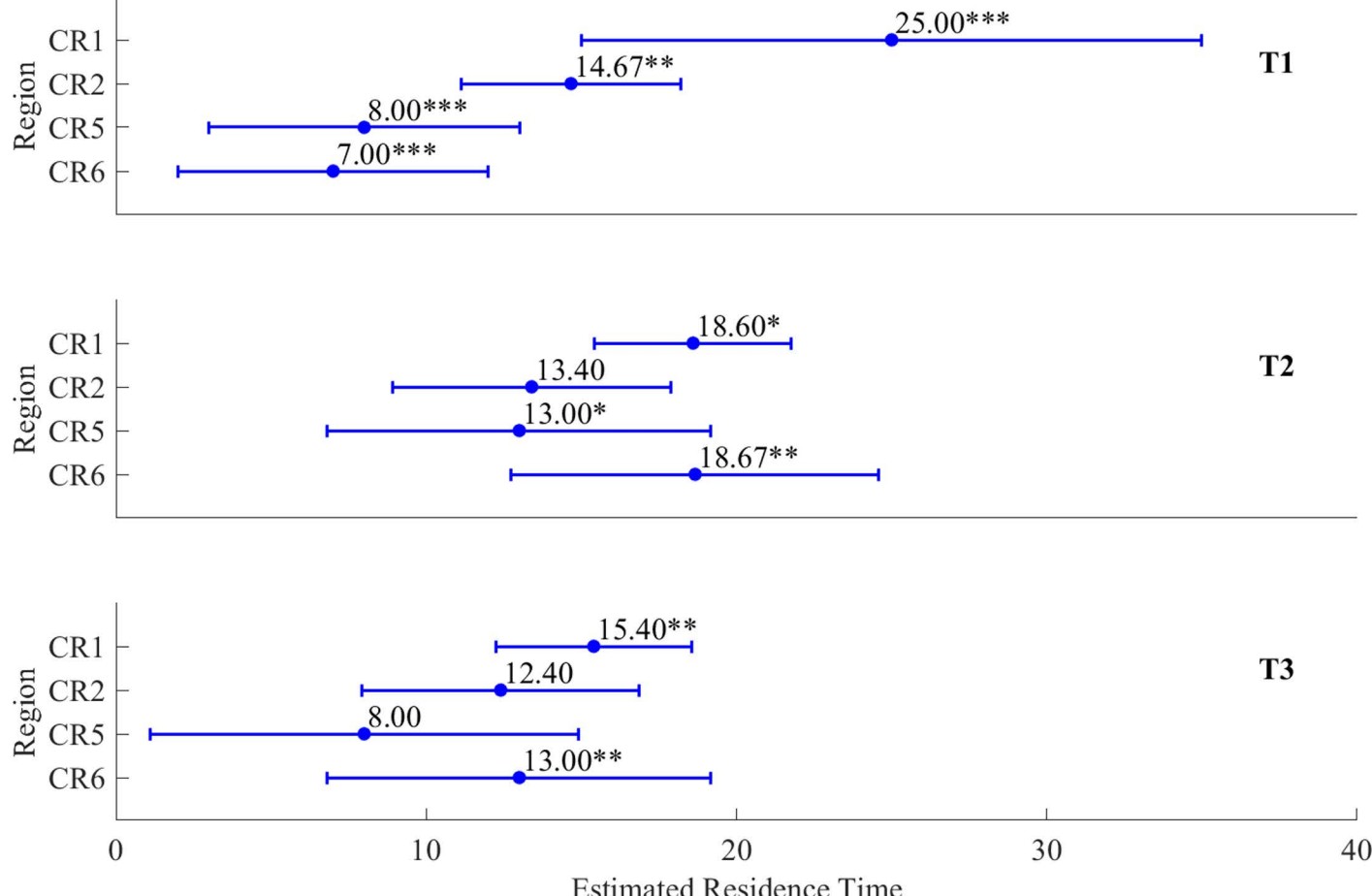

**Fig 6. Comparison of differences in coastal residence time of Chum between regions (CR1, CR2, CR5, CR6) by SST types (T1–T3) based on a linear mixed-effects model.** The blue circles with error bars indicate the estimated residence time of Chum. '*' indicates that $p < 0.01$. '**' indicates that $p < 0.05$. '***' indicates that $p < 0.001$.

rivers, and similarly, peaked at around 19–20 °C in the southern area rivers (Fig 7). In type T3, the riverine migration volume increased at 18–19 °C in the northern rivers and peaked at around 19–20 °C in the southern rivers (Fig 7).

## Discussion

In the ESCK, SST influenced both the coastal arrival timing and residence time of Chum. The Chum was first caught in the northern area and then gradually through the southern area. Additionally, while the coastal arrival timing and residence time of the Chum varied according to the SST distribution type in each region, the migration patterns derived from the regions in which they were first caught remained unchanged. This suggests that Chum migrates progressively from the northern to the southern area.

The timing of Chum coastal arrival was advanced or delayed to a similar scale across all regions of the ESCK depending on the SST distribution type. There were no significant differences between types T1 and T3, and the entire ESCK cooled down either early or late. This is probably because the consistent cold water in the deep layers of the ESCK minimized the thermal challenges for Chum migrating southward, resulting in a relatively stable migration pattern.

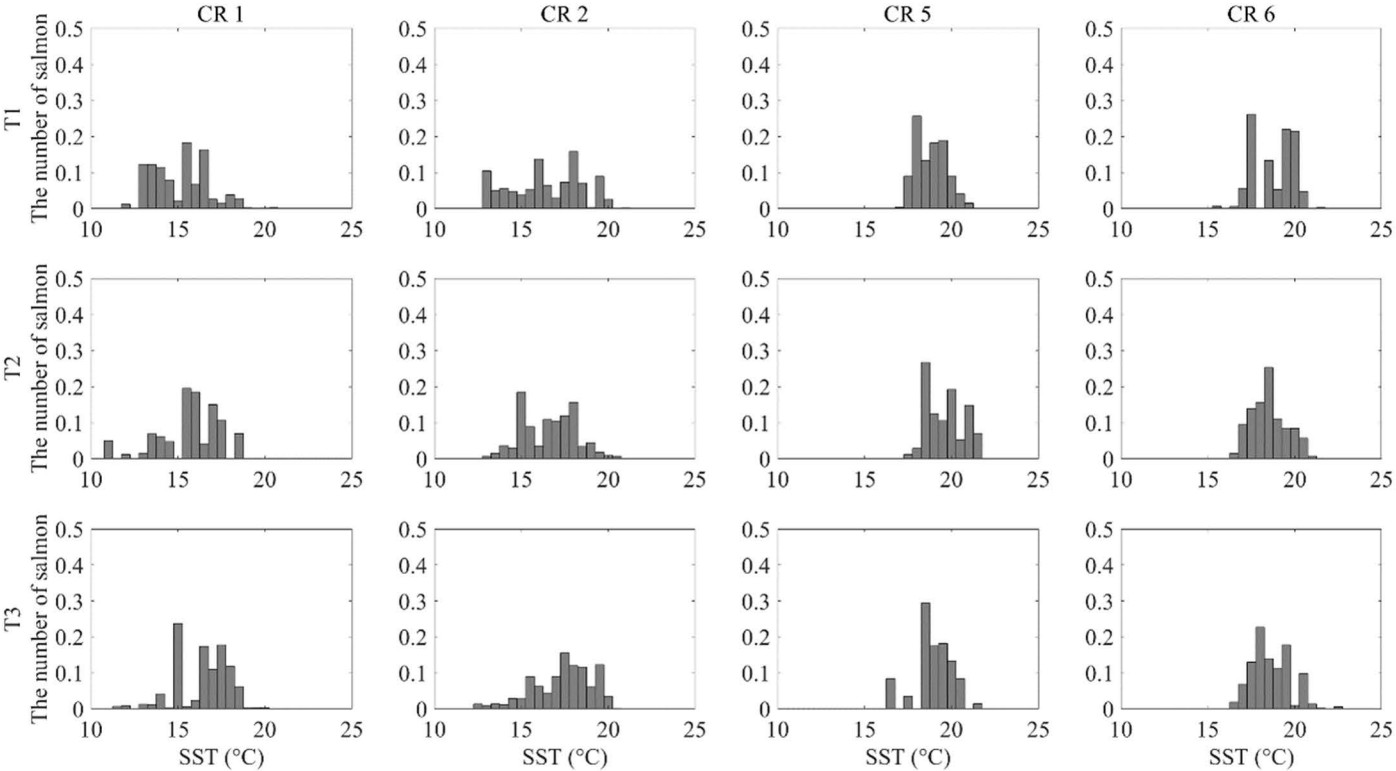

**Fig 7. The riverine migration volume of Chum adjacent to the ESCK (CR1, CR2, CR5, and CR6) in each water temperature range by the sea surface temperature distribution type (T1–T3).** The x-axis represents the water temperature, while the y-axis represents the riverine migration volume of Chum adjacent rivers within that water temperature range.

However, in type T2, the northern area cooled early, the southern area cooled late, and the coastal arrival timing was significantly earlier by approximately 4 days compared to T1 and T3 types across all regions of the ESCK. This is likely due to a Chum migration strategy to avoid abrupt temperature changes during spawning migration in environments with significant temperature differences, such as those between the northern and southern area under these conditions.

During their spawning migration, Chum may die from heat shock or develop gonadal disorders when they encounter radical changes in water temperature [58–60]. To mitigate this risk, Chum adopt a variety of strategies under radical environmental changes, such as staying longer in consistently cold deep layers or moving rapidly from the coastal waters to the river [27,28,37,61]. A subpolar front is distributed between approximately 37°–38° N in the ESCK, resulting in rapid vertical and horizontal temperature changes [46,62]. Chum returning to spawn in the ESCK primarily migrate southward, originating from northern areas [53]. In the ESCK, Chum in the northern area are unlikely to detect the environmental conditions in the distant southern area, stop their southward migration, and move into northern rivers. Similarly, under T2 conditions, it is improbable for Chum to reverse their migration northward after encountering the thermal boundary where cold and warm water masses meet. Instead, our findings suggest an alternative behavioral strategy. As Chum approaches the thermal boundary, they may adopt one of two responses: (a) continue migrating southward rapidly, enduring the stress of high temperatures and abrupt thermal changes, reach their natal river, or (b) halt their southward migration near the subpolar front and move into non-natal northern rivers adjacent to this thermal boundary.

Under T2 conditions in the ESCK, if the Chum adopt behavioral strategy (a), described above, coastal arrival migration will be concentrated in environments akin to T1 in the north and T3 in the south, as they migrate southward, regardless of the SST distribution type. However, if strategy (b) is adopted, some Chum that would have migrated to the southern area may instead move to the northern area adjacent the thermal boundary when the upper layers of the southern area are warm. This would lead to an increased concentration of Chum migration in high-temperature environments in the northern area compared to the patterns under conditions similar to T1. Subsequently, the number of Chum moving to the southern area would decrease when the upper layer of the southern area became cold, resulting in a concentrated increase in coastal arrival migration in high-temperature environments in both the southern and northern area.

When the ESCK is under T2 conditions, the volume of migrating Chum arriving in coastal waters in the north adjacent the thermal boundary (such as CR3) will be concentrated in high-temperature environments (about 18–20 °C), unlike the situation under T1 conditions. In the south adjacent the thermal boundary (such as CR4), the migration volume will be concentrated in high-temperature environments (20–21 °C), unlike the situation under T3 conditions. This suggests that under T2 conditions, some of the Chum heading south to spawn chose to move to northern rivers adjacent the thermal boundary instead to reduce the stress caused by radical changes in water temperature and to increase their spawning success. According to this strategy, under T2 conditions, the return rate of Chum to northern rivers adjacent the thermal boundary would be higher than that in an average year, while the return rate to southern rivers would be lower. The strategy in which some Chum enter to non-natal rivers affects genetic diversity, distribution changes, ecosystem stability, resource management, and evolutionary perspectives [31,60–64]. This includes facilitating genetic exchanges between populations, pioneering new habitats, and maintaining ecosystem stability [34,63–67].

If, in the future, we study differences in Chum movement and behavior by SST distribution type focusing on the return rate rather than migration timing, we can better understand Chum climate change responses and adaptation strategies. This approach would provide useful data for predicting Chum distribution and resource fluctuations in the North Pacific.

Chum spent less time in the coastal waters of the southern area compared to the northern one. Although the overall pattern of coastal residence time between the northern and southern areas remained consistent, the direction and extent of change (shortening or extending) varied depending on the SST distribution type. Coastal residence time in the northern area was shorter under T3 compared to T1 or T2, while it was longer in the southern area. In environments where Chum primarily entered rivers, the coastal residence time in the northern area progressively decreased with higher temperatures from T1 to T2 and from T2 to T3. In contrast, in the southern area, the coastal residence time remained relatively stable within a consistent temperature range of approximately 19–20 °C. This efficient adjustment of coastal residence time by Chum in the ESCK, where various environmental conditions present, is believed to reflect their behavioral strategies to align their river entry timing with optimal reproductive maturation at their spawning grounds.

Chum metabolism accelerates, and energy is quickly consumed, in high-temperature environments [23,37,68]. Because Chum rarely feed during spawning migration, the presence of high-temperature water in the upper layer during their migration increases the time they stay in the deep cold-water layers and the frequency of their vertical movement [69–71]. Chum returning to the northern rivers may travel relatively short distances compared to those returning to the southern rivers, expend less energy as they move through cold waters, and arriving in the coastal waters while still immature [53,72]. In contrast, Chum returning to southern rivers travel further and, as they move through warmer waters, expend more energy, arriving at coastal waters in an imminent spawning state [53,72].

As coastal waters cool more rapidly, Chum returning to northern rivers may benefit from staying in the coastal waters longer to increase their metabolism, rather than migrating directly to the rivers [28,32,68,73]. This is because these rivers are usually colder than the coastal waters during the spawning season. Chum returning to the southern rivers expend a substantial amount of energy and arrive in the coastal waters in an imminent spawning state; therefore, it may be advantageous for them to move quickly to the rivers shortly after the upper layers cool down [28,32,68,73]. For this reason, Chum returning to northern rivers and arriving at the coast in an immature state are likely to spend less time in the coastal waters, as prolonged high temperatures in the upper coastal layers accelerate their gonadal development. Conversely, Chum returning to southern rivers, despite having mature gonads, are likely to spend more time in the coastal waters owing to the challenging conditions in coastal waters.

In this study, we could not analyze the gonadal maturity of the Chum in each of the coastal water and river regions. However, future studies comparing the maturity of Chum upon their arrival in coastal waters and immediately after entering the rivers in different regions will provide a deeper understanding of their response strategies to the distribution of warm water masses in the northern and southern area.

Our findings provide significant implications for the development of resource management and conservation strategies for Chum salmon. For example, the differences in migration patterns according to SST distribution types could be utilized to adjust fishing schedules or implement fishing bans in specific areas. Currently, a uniform fishing ban period is applied across all regions, regardless of regional differences in migration timing or environmental conditions. However, our findings suggest that fishing ban periods could be adjusted based on environmental patterns and the unique characteristics of each region. Considering the observed variations in migration strategies between the northern and southern regions, it is crucial to establish region-specific management strategies that address the unique environmental conditions and behavioral responses of Chum salmon in each area. By tailoring conservation measures to regional dynamics, resource sustainability and population resilience can be better ensured.

As global warming, Chum resource management is shifting from increasing release volumes to enhancing early-life survival rates of Chum [74–77]. Historically, Chum releases followed fixed schedules, disregarding coastal environmental variability, but future strategies should instead optimize release timing based on regional conditions and SST patterns. Expanding studies to include early-life stages would provide valuable insights for sustainable management and ecosystem health.

## Conclusion

In the ESCK, which represents the southernmost distribution of spawning Chum, Chum migrate progressively southward from the northern area, and their coastal arrival timing and residence time vary according to SST distribution. In the ESCK, the temperature difference between the northern and southern areas was a significant environmental factor that influenced the arrival timing of Chum. The arrival time was accelerated when the temperature difference between these regions increased. In rivers adjacent to the ESCK, the distribution period of warm water in the upper coastal layers near the rivers is an important environmental factor affecting the coastal residence time of Chum. The longer the high SST persisted, the shorter the coastal residence time of Chum in the northern area, whereas their coastal residence time increased in the southern area. These findings enhance our understanding of how the poleward shift in warm water masses associated with climate change influences salmon migration and behavioral patterns.

## Supporting information

**S1 Table. Years corresponding to each coastal seawater temperature distribution type.**
(DOCX)

**S2 Table. AIC results for each nonlinear regression equation (polynomial, exponential, and logistic) between Chum coastal arrival timing and coastal sea surface water temperature by type of sea surface water temperature distribution and region.**
(DOCX)

**S3 Table. AIC results for each nonlinear regression equation (polynomial, exponential, and logistic) between the coastal residence time and coastal sea surface water temperature by type of sea surface water temperature distribution and region.**
(DOCX)

**S4 Table. Results of the polynomial regression model between Chum coastal arrival timing and coastal sea surface water temperature by region and type of sea surface water temperature distribution.**
(DOCX)

**S5 Table. Results of the polynomial regression model between coastal residence time of Chum and coastal sea surface water temperature by region and type of sea surface water temperature distribution.**
(DOCX)

**S1 Fig. Long-term variations of annual mean vertical temperature distribution in each region (CR1–CR6).** The x-axis represents the years, the y-axis represents the depth, and the colors indicate temperature. The temperature range is set to 14–20 °C, which corresponds to the optimal temperature range for Chum migration during the spawning season in the ESCK.
(TIF)

**S2 Fig. The number of Chum salmon returning to the ESCK.**
(TIF)

**S3 Fig. Mean of sea surface water temperature by date for each region (CR1–CR6) according to the types of sea surface water temperature distribution (T1–T3) in the ESCK.**
(TIF)

**S4 Fig. The estimated median date of migration of the Chum arriving at the ESCK (CR1–CR6) and entering adjacent rivers.** Black dots represent coastal migration timing, and red dots indicate river entry timing.
(TIF)

## Acknowledgments

The authors would like to express their sincere gratitude to the Korea Fisheries Resources Agency, Taehwa River Eco Center, Gyeongsangnam-do Research Center for Freshwater Fish, and the National Federation of Fisheries Cooperatives for providing invaluable data on the daily salmon catch volumes.

## Author contributions

**Conceptualization:** Beom-Sik Kim, Chung Il Lee.

**Data curation:** Beom-Sik Kim, Ju Kyoung Kim, Chung Il Lee.

**Formal analysis:** Beom-Sik Kim.

**Funding acquisition:** Beom-Sik Kim, Hae Kun Jung, Chung Il Lee.

**Methodology:** Beom-Sik Kim, Chung Il Lee.

**Software:** Beom-Sik Kim.

**Supervision:** Chung Il Lee.

**Visualization:** Beom-Sik Kim, Jong Won Park.

**Writing – original draft:** Beom-Sik Kim, Chung Il Lee.

**Writing – review & editing:** Hae Kun Jung, Jong Won Park.

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
