## [Decision Letter · Decision Letter 0]

13 Nov 2024

PONE-D-24-40631Temporal distribution shifts of Chum salmon (Oncorhynchus keta) with sea surface temperature changes at their southern limit in the North PacificPLOS ONE

Dear Dr. Lee,

Thank you for submitting your manuscript to PLOS ONE. After careful consideration, we feel that it has merit but does not fully meet PLOS ONE’s publication criteria as it currently stands. Therefore, we invite you to submit a revised version of the manuscript that addresses the points raised during the review process.

We look forward to receiving your revised manuscript.

Kind regards,

Abdul Azeez Pokkathappada, Ph.D.

Academic Editor

PLOS ONE

3. Thank you for stating the following financial disclosure: [This research was supported by Basic Science Research Program through the National Research Foundation of Korea(NRF) funded by the Ministry of Education(RS-2024-00412976), and the National Institute of Fisheries Science, Ministry of Oceans and Fisheries(R2024008), and the Korea Institute of Marine Science & Technology Promotion (KIMST) funded by the Ministry of Oceans and Fisheries (20220558).]. Please state what role the funders took in the study. If the funders had no role, please state: "The funders had no role in study design, data collection and analysis, decision to publish, or preparation of the manuscript." If this statement is not correct you must amend it as needed. Please include this amended Role of Funder statement in your cover letter; we will change the online submission form on your behalf.

Additional Editor Comments:

The study on Chum salmon (Oncorhynchus keta) distribution shifts in response to sea surface temperature (SST) changes is highly relevant to current marine ecological research. However, the manuscript requires significant revisions to meet publication standards, particularly in terms of experimental rigor, statistical analysis, and clarity in presenting conclusions. One key area for improvement is the framing of the research question. By refining and more clearly defining the objectives, the authors can better select appropriate statistical methods that align with their hypothesis, resulting in a stronger and more cohesive analysis. A well-defined hypothesis will help focus the statistical approach and improve the alignment of results with the study’s goals.

Moreover, while SST is a critical factor influencing marine species distribution, other environmental variables such as ocean currents, salinity, and nutrient availability also play a substantial role in shaping the migration patterns of Chum salmon. The current study appears to treat SST as a standalone variable, which could oversimplify the complex interactions affecting salmon behavior. Expanding the analysis to incorporate these confounding variables, either through additional data collection or by referencing existing data, would provide a more comprehensive understanding of the drivers behind distribution shifts. If including these factors is not feasible, the authors should explicitly acknowledge them as limitations in the discussion to provide context for their findings.

Lastly, the conclusions could benefit from clearer presentation and stronger linkage to the study's objectives and data. Revisiting the narrative structure of the results section, with an emphasis on clarity and data relevance, would help ensure that the conclusions are well-supported by the data and leave a lasting impact. Additionally, visual aids such as multi-layered mapping or comparison charts could enhance the interpretation of results, making the observed relationships easier to understand. With these revisions, the study would be better positioned to offer a robust and scientifically sound analysis of the factors influencing Chum salmon migration patterns.

Reviewers' comments:

Reviewer's Responses to Questions

**Comments to the Author**

1. Is the manuscript technically sound, and do the data support the conclusions?

Reviewer #1: Partly

Reviewer #2: Yes

2. Has the statistical analysis been performed appropriately and rigorously? 

Reviewer #1: I Don't Know

Reviewer #2: Yes

3. Have the authors made all data underlying the findings in their manuscript fully available?

Reviewer #1: No

Reviewer #2: Yes

4. Is the manuscript presented in an intelligible fashion and written in standard English?

Reviewer #1: No

Reviewer #2: Yes

5. Review Comments to the Author

Reviewer #1: The authors explore differences between coastal and river migration timing for multiple Chum salmon river systems along the southeastern coast of Korea. They observed differences in coastal migration timing as a function of the difference in SST between northern and southern areas during the period of return migration, as well as differences in river timing as a function of regional temperature conditions. Understanding the potential for behavioural adaption as a result of changing climate is of particular interest for populations such as ESCK Chum salmon, which exist at the southern end of their range.

While the authors pose an interesting research question, the conclusions drawn from their results seemed hypothetical and not always supported by their findings. I also found the level of description with respect to the data sources in the methods to be lacking, which contributed to some confusion in interpreting the results. The display of the results seem inconsistent between the coastal and river timing analyses, with different formatting and labelling applied to similar figures. I also had some concerns with respect to the appropriateness of the statistical analysis and feel like the authors could have investigated the key relationships underlying sea surface temperature and return migration timing using a more comprehensive approach.

Although the research topic is certainly relevant and the results show support for a relationship between ocean temperature conditions and Chum salmon migration behaviour, without major revisions I do not feel that the current article merits publication, in particular due to failure to meet standards related to “Experiments, statistics, and other analyses are performed to a high technical standard and are described in sufficient detail” and “Conclusions are presented in an appropriate fashion and are supported by the data”. More clearly framing their research question may help the authors better refine their choice of statistical analysis and presentation of results to better support their original hypothesis.

For more detail regarding my decision and suggestions for major and minor revisions, please see below.

1. The study presents the results of original research.

While the authors acknowledge previous studies had shown a relationship between sea surface temperature and Korean Chum salmon migration, to the best of my knowledge, the exact type of relationships in the current publication have not been reported elsewhere.

2. Results reported have not been published elsewhere.

Not to my knowledge.

3. Experiments, statistics, and other analyses are performed to a high technical standard and are described in sufficient detail.

I found the description of the underlying data sources in the paper to be lacking. In particular, it was unclear the location, number, and time period over which sea surface temperature data was collected. In addition, it would useful for the authors to address underlying assumptions resulting from the use of catch data as a proxy for abundance. For example, how did they standardise for changes in effort or catchability? I also found it strange that the river locations were referred to generically instead of actually naming, and providing some basics characteristics of, each system. For example, what is the relative return size of the Chum salmon population to each river? Are these all wild stocks? Hatchery? Or a combination? A figure showing historical coastal and river entry timing (e.g. year vs. timing) for each location would have also provided a useful baseline for understanding the underlying differences across locations, and across different sea surface temperature SST “types”.

While the use of SST “types” was interesting, I was still not clear as to how this particular characterisation of ocean conditions would be preferable to using a numerical indicator such as SST (or SST anomaly) averaged over a relevant time period (e.g. centred around peak historical migration timing). The authors do not clearly characterise how they defined a year as cooling faster or slower than average. I was also concerned about the limited sample size (3-years only) for “Type 1” conditions, and the fact that the timeseries ended in 2018. If the author’s want to continue with the current SST “types” as defined in the current study, I think they need to:

(a) better justify their approach for defining alternative SST “types”

(b) more clearly explain how the metrics to define each “type” was calculated

(c) consider the trade-offs between using a categorical variable as opposed to a numerical representation of ocean temperature conditions

I was also unclear on the author’s definition of river entry timing. On L. 180-181, the authors state: “The river entry timing in each region was determined as the period between the MDMT of the ESCK and that of the adjacent river”. I interpreted this to mean that the river timing was being represented as the relative difference between arrival in coastal areas and arrival into the river. I presume this is because river entry timing is a function of both coastal arrival timing AND how long fish delay in coastal areas before entering the river, and the authors were more interested in the latter. i.e. assess whether fish were delaying longer in coastal areas before migrating upstream. However, subsequent wording in the body of the report and in the figures seemed to suggest that the analyses were performed directly on the median arrival timing into the river.

Given that a large part of the focus on the results was on latitudinal differences in responses to changing temperature, particularly for river timing, I also wonder if the authors considered using the latitude of each river mouth as a continuous variable as opposed to the fixed location definitions.

The author’s used AIC to select across fixed model types. Another option would have been to apply a step-wise AIC to the most complex model form. Given the small sample sizes in some groupings (e.g. only 3 years of data for SST Type “1”; only 4 locations used for river timing analysis) and the appearance of unequal variances in some of the figures, I wonder if the authors considered any failure of underlying model assumptions for their linear analysis. I also did not see any major advantage in the non-linear analysis in terms of supporting the authors results. The main question the authors appeared to be asking is if the peak river migration varied as a function of SST. This could be done much more simply, and potentially grouped together with the linear analysis by including SST associated with the median timing date as a variable in the model.

I did not find the non-linear analysis to provide a lot of additional useful information. My main interpretation of the resulting figures was that the peak migration tends to be associated with a narrow temperature range for a given river system, with coastal and river timing for northern systems being associated with cooler SSTs than for southern systems. I think this relationship could be illustrated more simply by plotting median timing vs. SST associated with median timing. The histograms are useful in that they illustrate that there is some capacity for adaptation in that the salmon do enter the coastal or river systems over a range of temperatures. But maybe the authors could include a vertical line illustrating the peak timing of migration with a label showing the associated SST. The AIC values for the northern rivers also appeared to be much higher, on average, than for the southern rivers which also raises the question of how useful this analysis was for the northern systems.

4. Conclusions are presented in an appropriate fashion and are supported by the data.

There was some apparent inconsistency in the presentation of figures throughout the report, with differences in labelling and formatting between figures describing results for coastal vs. river temperatures. While the focus was on trends, and so relative differences are important, the current presentation masks the underlying differences in timing between northern and southern sites, which is referenced by the authors in the text, but would be more clear if the actual timing was displayed.

I found it challenging to link the patterns of earlier coastal arrival timing in both the north and south under Type T2 conditions to the explanations raised by the authors in the discussion. Some specific examples are provided below.

L. 345 – 348. The authors state that earlier coastal timing under the type T2 SST scenario was because the Chum salmon are trying to avoid migration through abrupt temperature changes. However, it is unclear how southern temperatures would impact the timing into northern coastal areas if the fish migrate from the north into the south. If southern migrants strayed into northern rivers to avoid higher temperatures in the south, then potentially this could impact both abundance and timing in the north but the results are not produced in a clear way to support this argument.

L. 374-378. The authors discuss a change in fish abundance in the north vs. the south under type T2 SST conditions, but no analysis was shown to support this result.

L. 391-393. Authors state that timing into northern rivers was associated with higher temperatures as overall cooling patterns shifted from earlier to later. However, this does not seem supported by the results in Table S5, where the relationship isn’t significant for CR1 or CR2 under T1, or in CR1 under T2.

The authors could have expanded more on how behavioural strategies between northern and southern systems may have been impacted because southern migrants are already dealing with warmer temperatures on average which are closer to the physiological limits for salmon than the conditions experienced by migrants in the north. It would also have been informative to tie the results and discussion back to statements in the introduction relating to improved returns to the ECSK even as environmental conditions have become less favourable. Is this because of adaptability? Or have there also been coincident declines in fisheries harvest contributing to this trend?

5. The article is presented in an intelligible fashion and is written in standard English.

Paper was difficult to follow in sections due to unclear definitions, changes in flow between methods and results, and inconsistency in figure labelling. For example, the term “region” was sometimes used to describe individual rivers and at other times to describe the northern/southern aggregations of study sites. The term “advanced” timing could also be interpreted as fish arriving either earlier, or later, to a system.

6. The research meets all applicable standards for the ethics of experimentation and research integrity.

NA

7. The article adheres to appropriate reporting guidelines and community standards for data availability.

NA

Reviewer #2: The article on the temporal distribution shifts of Chum salmon (Oncorhynchus keta) in relation to sea surface temperature (SST) changes provides valuable insights into how marine species respond to climate change. However, several aspects could be strengthened to enhance the study’s rigor and its broader implications.

The focus on analyzing the impact of SST on Chum salmon migration is timely and relevant, yet the study appears to treat SST as a standalone variable without sufficiently considering other influential factors. For instance, changes in ocean currents, salinity, and nutrient availability are critical factors that can also impact the distribution and migration patterns of marine species. By not addressing these potential confounding variables, the study may oversimplify the complex interplay of factors affecting Chum salmon behavior.

I recommend modifying and updating the first statement of the introduction with given studies as [1,2] “Similar to agricultural ecosystems, climate change also impacts marine ecosystems [3]. Recently, marine ecosystems have been affected…

[1] https://doi.org/10.1016/j.agsy.2024.103994

[2] https://doi.org/10.3389/fenvs.2021.826838

[3] https://doi.org/10.1016/B978-0-12-822373-4.00024-0

The categorization of SST into three types (T1–T3) provides a structured approach to examining temperature variations. However, the definition of these categories appears somewhat arbitrary, with limited justification for why these specific types were chosen. It would be beneficial to clarify the criteria used to distinguish between rapid and slow cooling, especially given the potential variability in regional oceanic conditions.

While the study identifies a temporal shift in riverine entry timing based on SST changes, the conclusions could be more robust if supported by additional ecological data. For instance, the study could benefit from integrating biological factors such as the age structure of the salmon population, prey availability, and predator pressure, which might influence the observed migration patterns.

The lack of significant differences in coastal arrival timing between the T1 and T3 scenarios raises questions about the sensitivity of the analysis. The study could delve deeper into why these changes in SST, despite being categorized as rapid or slow cooling, did not produce observable effects on salmon arrival. A discussion on the potential buffering mechanisms that Chum salmon might possess against temperature fluctuations would provide a more comprehensive understanding.

The article presents its results as if they are directly applicable to predicting future climate change impacts on Chum salmon distribution. However, it does not account for the fact that SST is only one of many factors affected by global warming. The study's findings would be more meaningful if contextualized within broader climate change projections, such as ocean acidification or changes in seasonal precipitation patterns, which can also alter habitat suitability for salmon.

Additionally, while the study claims that the findings are useful for predicting changes in Chum distribution and resources in the North Pacific, it does not provide specific recommendations for fisheries management or conservation strategies. This is a missed opportunity to connect the research to actionable policy measures, especially given the importance of Chum salmon for both ecological systems and coastal communities.

Lastly, the article could be improved by expanding its geographical scope. Focusing solely on the eastern and southern coastal waters of Korea limits the generalizability of the results. Including data from other parts of the North Pacific could provide a more holistic view of the species’ adaptation strategies across their entire southern distribution range. some regional studies are suggested to incorporate and validate with your results in discussion section.

https://doi.org/10.3390/atmos14010079

https://doi.org/10.1088/1748-9326/ac8fa6

https://doi.org/10.3389/fenvs.2023.1228817

https://doi.org/10.3390/w15132420

6. PLOS authors have the option to publish the peer review history of their article (what does this mean? ). If published, this will include your full peer review and any attached files.

**Do you want your identity to be public for this peer review?** For information about this choice, including consent withdrawal, please see our Privacy Policy .

Reviewer #1: No

Reviewer #2: No

---

## [Author Response · Author response to Decision Letter 1]

30 Nov 2024

24 Nov 2024

Abdul Azeez Pokkathappada, Ph.D.

Academic Editor

PLOS ONE

Dear Editor:

We wish to re-submit the revised manuscript titled “Temporal distribution shifts of Chum salmon (Oncorhynchus keta) with sea surface temperature changes at their southern limit in the North Pacific” The manuscript number is PONE-D-24-40631. This revision and re-submitted manuscript were prepared with Beom-Sik Kim, Hae Kun Jung, Jong Won Park, and Ju Kyoung Kim, with all authors ensuring alignment with the reviewers’ suggestions and the highest quality.

We sincerely appreciate you for taking the time to review our manuscript. Your insightful feedback and suggestions have been incredibly helpful in refining and strengthening our study. The attached response letter includes detailed answers to all reviewer comments and descriptions of the corresponding revisions. Reviewer comments are provided in black text, and our responses are highlighted in blue.

We hope this revised manuscript meets the high standards of your journal and look forward to your positive consideration.

Sincerely,

Chung Il Lee

Department of Marine Ecology and Environment

Gangneung-Wonju National University

Gangneung 25457, Republic of Korea

Tel. +82-33-640-2855

E-mail: leeci@gwnu.ac.kr

Dear Editor and Reviewers,

We sincerely appreciate the time and effort you have dedicated to reviewing our manuscript, as well as the insightful feedback and suggestions you provided. Your comments have been invaluable in enhancing the depth and quality of our research.

We carefully reviewed all comments and have made substantial revisions to the manuscript. Specifically, we clarified the research questions and hypotheses, elaborated on the methods and materials, and restructured the figures and tables to enhance clarity. Additionally, we reinforced the ecological and resource management implications of our findings in the Discussion section.

We addressed the reviewers’ suggestions, including adding references, refining sentences, and aligning the study's focus with its scope.

Reviewer #1

Overview Comments:

The authors explore differences between coastal and river migration timing for multiple Chum salmon river systems along the southeastern coast of Korea. They observed differences in coastal migration timing as a function of the difference in SST between northern and southern areas during the period of return migration, as well as differences in river timing as a function of regional temperature conditions. Understanding the potential for behavioural adaption as a result of changing climate is of particular interest for populations such as ESCK Chum salmon, which exist at the southern end of their range.

While the authors pose an interesting research question, the conclusions drawn from their results seemed hypothetical and not always supported by their findings. I also found the level of description with respect to the data sources in the methods to be lacking, which contributed to some confusion in interpreting the results. The display of the results seem inconsistent between the coastal and river timing analyses, with different formatting and labelling applied to similar figures. I also had some concerns with respect to the appropriateness of the statistical analysis and feel like the authors could have investigated the key relationships underlying sea surface temperature and return migration timing using a more comprehensive approach.

Although the research topic is certainly relevant and the results show support for a relationship between ocean temperature conditions and Chum salmon migration behaviour, without major revisions I do not feel that the current article merits publication, in particular due to failure to meet standards related to “Experiments, statistics, and other analyses are performed to a high technical standard and are described in sufficient detail” and “Conclusions are presented in an appropriate fashion and are supported by the data”. More clearly framing their research question may help the authors better refine their choice of statistical analysis and presentation of results to better support their original hypothesis.

For more detail regarding my decision and suggestions for major and minor revisions, please see below.

Answer:

We appreciate you for taking the time to review our study and for providing invaluable feedback that has greatly contributed to improving our research. We understand the concerns raised by the reviewer, including the lack of clarity in the research questions, insufficient explanation of data sources, inconsistencies in the presentation of results, and the issues regarding the statistical analysis approach. Based on this feedback, we have made significant revisions to address these issues.

First, to improve the clarity of the research questions, we revised the introduction and objectives sections to clearly present the hypotheses and goals of our study. This has enhanced the logical flow and analytical direction of the manuscript. Additionally, we provided more detailed descriptions of the sea surface temperature (SST) and catch data, including their collection methods and spatial and temporal scope, to strengthen the explanation of the data sources.

In the results section, inconsistencies in the labeling and formatting of visual materials have been addressed. This difference was caused by the different LME model equations that are best for analyzing the difference between coastal arrival times and coastal stay times by SST type. We revised the methods and results sections to clarify these differences and improve reader comprehension.

Regarding the statistical analysis, we re-evaluated the statistical approaches to ensure they appropriately support the main hypotheses of the study. We thoroughly validated the model assumptions and confirmed their suitability. Furthermore, we clarified the limitations of the analytical methods in the discussion section and proposed directions for future research.

Once again, we sincerely appreciate the valuable advice and feedback. We hope these improvements have contributed to significantly advancing the manuscript. Detailed answers to specific questions are provided below.

Q1) I found the description of the underlying data sources in the paper to be lacking. In particular, it was unclear the location, number, and time period over which sea surface temperature data was collected. In addition, it would useful for the authors to address underlying assumptions resulting from the use of catch data as a proxy for abundance. For example, how did they standardise for changes in effort or catchability? I also found it strange that the river locations were referred to generically instead of actually naming, and providing some basics characteristics of, each system. For example, what is the relative return size of the Chum salmon population to each river? Are these all wild stocks? Hatchery? Or a combination? A figure showing historical coastal and river entry timing (e.g. year vs. timing) for each location would have also provided a useful baseline for understanding the underlying differences across locations, and across different sea surface temperature SST “types”.

A1) Thank you for your valuable feedback. In response to your suggestions, we have added detailed explanations regarding the sea surface temperature (SST) and catch data in the Materials and Methods section (L133–138, L144–146, L159–164). The rivers CR1–CR6 are areas where Chum are annually caught and released as part of hatchery programs. However, we cannot determine the exact proportion of individuals returning to each river that originate from hatcheries versus those that are wild. Additionally, we have included year-by-year data on the MDMT of Chum arriving at coastal waters and entering rivers in S4 Fig. We sincerely appreciate your insightful comments, which have greatly contributed to improving our study. Thank you again.

Q2) While the use of SST “types” was interesting, I was still not clear as to how this particular characterisation of ocean conditions would be preferable to using a numerical indicator such as SST (or SST anomaly) averaged over a relevant time period (e.g. centred around peak historical migration timing). The authors do not clearly characterise how they defined a year as cooling faster or slower than average. I was also concerned about the limited sample size (3-years only) for “Type 1” conditions, and the fact that the timeseries ended in 2018. If the author’s want to continue with the current SST “types” as defined in the current study, I think they need to:

(a) better justify their approach for defining alternative SST “types”

(b) more clearly explain how the metrics to define each “type” was calculated

(c) consider the trade-offs between using a categorical variable as opposed to a numerical representation of ocean temperature conditions

A2) Thank you for your valuable feedback, which has greatly contributed to improving this study. The ESCK is characterized by the interaction of cold-water masses flowing southward from the north and warm water masses moving northward from the south, resulting in distinct SST distribution types along the coastline. As you mentioned, studying the relationship between SST and Chum migration timing is an interesting research topic, and we have submitted related work to another journal, which is currently under review. In this study, we focused on investigating how Chum returning to Korea respond to these SST distribution types.

The SST types in this study were defined as follows: T1 and T3 represent conditions where the northern and southern areas cool below 20°C around the same time. However, T1 indicates an earlier-than-average cooling, while T3 represents a later-than-average cooling. T2 reflects conditions where the northern area cools faster than the southern area, regardless of whether the cooling is earlier or later than the average. These definitions have been clarified and revised in L164-174.

Q3) I was also unclear on the author’s definition of river entry timing. On L. 180-181, the authors state: “The river entry timing in each region was determined as the period between the MDMT of the ESCK and that of the adjacent river”. I interpreted this to mean that the river timing was being represented as the relative difference between arrival in coastal areas and arrival into the river. I presume this is because river entry timing is a function of both coastal arrival timing AND how long fish delay in coastal areas before entering the river, and the authors were more interested in the latter. i.e. assess whether fish were delaying longer in coastal areas before migrating upstream. However, subsequent wording in the body of the report and in the figures seemed to suggest that the analyses were performed directly on the median arrival timing into the river.

A3) Thank you for your valuable feedback. Based on your feedback, we have replaced the term "river entry timing" with "coastal residence time" throughout the manuscript (such as L197-198).

Q4) Given that a large part of the focus on the results was on latitudinal differences in responses to changing temperature, particularly for river timing, I also wonder if the authors considered using the latitude of each river mouth as a continuous variable as opposed to the fixed location definitions.

A4) Thank you for raising an important point. In this study, the "Regions" variable is defined based on latitude, from north to south, effectively reflecting changes in latitude. Additionally, we conducted supplementary analyses comparing the results of using latitude as a continuous variable versus the fixed location definitions (Regions). The results from both approaches were nearly identical, suggesting that the "Regions" variable sufficiently represents latitudinal variation and adequately explains the response patterns based on latitude, which is the main focus of this study. Therefore, we chose to retain the fixed location definitions in the final model to maintain clarity in interpretation.

Q5) The author’s used AIC to select across fixed model types. Another option would have been to apply a step-wise AIC to the most complex model form. Given the small sample sizes in some groupings (e.g. only 3 years of data for SST Type “1”; only 4 locations used for river timing analysis) and the appearance of unequal variances in some of the figures, I wonder if the authors considered any failure of underlying model assumptions for their linear analysis. I also did not see any major advantage in the non-linear analysis in terms of supporting the authors results. The main question the authors appeared to be asking is if the peak river migration varied as a function of SST. This could be done much more simply, and potentially grouped together with the linear analysis by including SST associated with the median timing date as a variable in the model.

A5) Thank you for your valuable feedback. This study designed to analyze the response of Chum to sea temperature distribution types from a temporal distribution perspective using a simple and clear manner. To achieve this, SST types were treated as categorical variables, focusing on their primary effects rather than exploring additional variables or complex interaction effects. We compared three model equations that included SST types and their interactions with regions, and the best model was selected based on AIC. Employing a stepwise AIC approach to analyze a more complex initial model was considered beyond the scope of this study, as it did not align with our research focus. This approach was chosen to clearly identify the effects of SST distribution types and fulfill the study’s objectives effectively.

Additionally, the interactions between SST types and regions were included as key explanatory variables in the analysis. The exclusion of more complex interactions or additional variables was an intentional decision to maintain consistency in study design and focus. While future research could explore these additional variables and expanded analyses, this study focused specifically on SST distribution types to ensure a clear understanding of their effects.

As you noted, the small sample size in some groups (e.g., three years of data for T1) is acknowledged. To address this, we utilized LME model and verified the normality and suitability during the analysis process, concluding that the model was appropriate for the data.

Q6) I did not find the non-linear analysis to provide a lot of additional useful information. My main interpretation of the resulting figures was that the peak migration tends to be associated with a narrow temperature range for a given river system, with coastal and river timing for northern systems being associated with cooler SSTs than for southern systems. I think this relationship could be illustrated more simply by plotting median timing vs. SST associated with median timing. The histograms are useful in that they illustrate that there is some capacity for adaptation in that the salmon do enter the coastal or river systems over a range of temperatures. But maybe the authors could include a vertical line illustrating the peak timing of migration with a label showing the associated SST. The AIC values for the northern rivers also appeared to be much higher, on average, than for the southern rivers which also raises the question of how useful this analysis was for the northern systems.

A6) Thank you for your valuable feedback. In this study, the nonlinear regression method was employed to compare the temperature ranges and catch patterns of Chum under different SST distribution types, providing insights into their behavioral strategies in coastal and river environments. Specifically, the analysis aimed to determine whether catches became more concentrated within specific temperature ranges as SST types progressed from T1 to T3, whether higher temperatures were uniquely associated with T2, or whether certain types exhibited evenly distributed catches across a variety of temperature ranges.

The nonlinear analysis revealed that the coastal arrival timing of Chum under T1 and T3 conditions was similar, while T2 showed ear

---

## [Decision Letter · Decision Letter 1]

8 Jan 2025

Temporal distribution shifts of Chum salmon (*Oncorhynchus keta* ) with sea surface temperature changes at their southern limit in the North Pacific

PONE-D-24-40631R1

Dear Dr. Lee,

We’re pleased to inform you that your manuscript has been judged scientifically suitable for publication and will be formally accepted for publication once it meets all outstanding technical requirements.

Kind regards,

Abdul Azeez Pokkathappada, Ph.D.

Academic Editor

PLOS ONE

Additional Editor Comments (optional):

Reviewers' comments:

Reviewer's Responses to Questions

**Comments to the Author**

1. If the authors have adequately addressed your comments raised in a previous round of review and you feel that this manuscript is now acceptable for publication, you may indicate that here to bypass the “Comments to the Author” section, enter your conflict of interest statement in the “Confidential to Editor” section, and submit your "Accept" recommendation.

Reviewer #2: (No Response)

2. Is the manuscript technically sound, and do the data support the conclusions?

Reviewer #2: (No Response)

3. Has the statistical analysis been performed appropriately and rigorously? 

Reviewer #2: (No Response)

4. Have the authors made all data underlying the findings in their manuscript fully available?

Reviewer #2: (No Response)

5. Is the manuscript presented in an intelligible fashion and written in standard English?

Reviewer #2: (No Response)

6. Review Comments to the Author

Reviewer #2: (No Response)

7. PLOS authors have the option to publish the peer review history of their article (what does this mean? ). If published, this will include your full peer review and any attached files.

**Do you want your identity to be public for this peer review?** For information about this choice, including consent withdrawal, please see our Privacy Policy .

Reviewer #2: No

---

## [Editor Report · Acceptance letter]

PONE-D-24-40631R1

PLOS ONE

Dear Dr. Lee,

I'm pleased to inform you that your manuscript has been deemed suitable for publication in PLOS ONE. Congratulations! Your manuscript is now being handed over to our production team.

Kind regards,

on behalf of

Dr. Abdul Azeez Pokkathappada

Academic Editor

PLOS ONE